# Revisiting the Concept of Stress in the Prognosis of Solid Tumors: A Role for Stress Granules Proteins?

**DOI:** 10.3390/cancers12092470

**Published:** 2020-09-01

**Authors:** Anaïs Aulas, Pascal Finetti, Shawn M. Lyons, François Bertucci, Daniel Birnbaum, Claire Acquaviva, Emilie Mamessier

**Affiliations:** 1Predictive Oncology Laboratory, Cancer Research Center of Marseille (CRCM), Inserm U1068, CNRS UMR7258, Institut Paoli-Calmettes, Aix Marseille Université, 13009 Marseille, France; FINETTIP@ipc.unicancer.fr (P.F.); BERTUCCIF@ipc.unicancer.fr (F.B.); daniel.birnbaum@inserm.fr (D.B.); claire.acquaviva@inserm.fr (C.A.); emilie.mamessier@inserm.fr (E.M.); 2Department of Biochemistry, Boston University School of Medicine, Boston, MA 02118, USA; sml433@nyu.edu; 3The Genome Science Institute, Boston University School of Medicine, Boston, MA 02118, USA

**Keywords:** stress granules, G3BP1, G3BP2, CAPRIN-1, USP10, TIA-1, TIAR, cancer prognosis, biomarker, metastasis, resistance, cell death, pro-survival properties

## Abstract

**Simple Summary:**

Stress Granules (SGs) were discovered in 1999 and while the first decade of research has focused on some fundamental questions, the field is now investigating their role in human pathogenesis. Since then, evidences of a link between SGs and cancerology are accumulating in vitro and in vivo. In this work we summarized the role of SGs proteins in cancer development and their prognostic values. We find that level of expression of protein involved in SGs formation (and not mRNA level) could serve a prognostic marker in cancer. With this review we strongly suggest that SGs (proteins) could be targets of choice to block cancer development and counteract resistance to improve patients care.

**Abstract:**

Cancer treatments are constantly evolving with new approaches to improve patient outcomes. Despite progresses, too many patients remain refractory to treatment due to either the development of resistance to therapeutic drugs and/or metastasis occurrence. Growing evidence suggests that these two barriers are due to transient survival mechanisms that are similar to those observed during stress response. We review the literature and current available open databases to study the potential role of stress response and, most particularly, the involvement of Stress Granules (proteins) in cancer. We propose that Stress Granule proteins may have prognostic value for patients.

## 1. Generalities

According to the World Health Organization, cancer is the second leading cause of death worldwide. In most cases, mortality is not due to the primary tumor itself, but to the occurrence of metastasis. At diagnosis, non-metastatic diseases are treated with first line systemic adjuvant (post-operative) and/or neoadjuvant (pre-operative) treatments in order to avoid tumor relapse by eradicating potential residual or micro-metastatic foci. In case of metastatic disease, the aim of systemic treatment is to achieve clinical remissions as durable as possible, which remains a palliative situation for most of solid cancers. In both situations, if the therapy does not eliminate all malignant cells, residual cancer cells may acquire migratory properties and develop drug resistance mechanisms, further contributing to difficult-to-treat metastasis [1].

While the number of drugs approved for cancer treatment increases, the number of complete remissions after first line treatment remains too low and metastases keep occurring because primary or secondary resistance develops. Different strategies are being developed in order to improve the results of systemic treatments. One of them is precision medicine which aims to overcome this situation by increasing the specificity of the administered treatment: DNA and RNA of patients’ metastases are now sequenced to identify genome alterations and determine the best therapeutic option after first line therapy based on actionable gene mutations and genome alterations. We are now entering the “one tumor at one time-point for one therapeutic strategy” era. However, this remains difficult in practice and the benefit remains to be proven on a large scale. Tumors are heterogeneous and metastases are the sum of years of genomic instability and individual environmental history. We might not have the ability to decipher this degree of complexity yet. However, in the meantime studies can identify new resistance pathways to increase efficiency of systemic treatments and improve patients’ remission. Growing evidence indicates that the malignant cell can respond to exposure to anti-cancer drugs through fast, “acute”, and “ready-to-use” biological components or processes [2]. For all these reasons, elements of the stress response could be an interesting path to explore in this context.

The stress response is an ancestral evolutionary mechanism acquired by the earliest cellular organisms to protect them from sudden environmental or intracellular changes [3]. At the cellular level, any sudden change in the environment that diverges from its optimal growth condition is considered as an insult that triggers a stress response. The stress response includes the activation of stress-responsive genes expression, such as those coding for heat shock proteins (hours to days’ time scale). Stress response also includes some really fast mechanisms (minutes to hours), such as the formation of “Stress Granules” (SGs) [4,5]. The fate of a stressed cell is ultimately determined by its ability to recover from the stress [6,7,8]. Actually, cancer cells face a diverse set of stress conditions during oncogenesis, such as hypoxia, nutrient deprivation, oxidative stress, and even anti-cancer drugs. They have all been reported to induce the formation of SGs in vitro [4,9,10,11,12,13,14,15,16]. It is surmised that SGs are transient triage centers that are designed to help cells to quickly and transiently modify translation to overcome stress and enhance survival, both of which could be used to escape cancer therapies [17,18,19].

In this review, we will focus on stress granules (SGs) as actors of cancer development. We will first start by presenting what SGs are and how they form, discuss their pro-survival effect and how they could be involved in cancer progression. Second, we will analyze the prognostic value of the major proteins that are involved in SGs formation. Finally, we will discuss the involvement and contribution of cellular stress responses to cancer evolution.

## 2. SGs Are Composed of Proteins Involved in the Regulation of mRNA Translation

Stress granules, first discovered in 1999, are membrane-less cytoplasmic condensates, visible by conventional and electron microscopy [4,20,21,22]. More recently, SGs have been show to handle like hydrogel structures in the cytoplasm and form what is called a liquid-liquid phase separation [23,24]. They have been observed in plants, yeast, worms, insects, and mammalian cells [4,25,26,27,28,29]. This high degree of conservation across species highlights their importance for cell survival and the maintenance of cell integrity [30]. These foci are composed of mRNAs, RNA-binding proteins, and 40S ribosomes [4,31]. It is generally admitted that the absence of a membrane ensures the rapid execution of SGs formation and the extreme lability and adaptability of their components (SGs’ proteins generally are present in cells in homeostatic conditions and only switch their localization and functions between basal and stress conditions). However, this has hindered the purification of these structures and the precise identification of SGs components by global analysis. Currently, even if methods have been reported to purify SGs markers [32,33], candidate approaches have been for a long time the only way to identify specific components. Most studies have used immunofluorescence and Florescent In Situ Hybridization (FISH) to identify proteins and mRNAs that are included in these structures. In 2015, an inventory of the literature indicated more than a hundred proteins recruited to SGs [31]. These proteins form an eclectic mix involved in various signaling pathways. Even if there is still no consensus to predict the recruitment of specific proteins to SGs, most of them interact with RNA or are involved in the metabolism of RNA. SGs are also composed of components that are involved in translation initiation, such as Eukaryotic Initiation Factor (EIF) 3 and EIF4 complex proteins or PABP (PolyA Binding Protein) [31]. The presence of these components is the consequence of a general translation inhibition that precedes SGs formation.

In homeostatic conditions, active translation is facilitated by the formation of a closed-loop mRNA during specific steps of translation [34] (Figure 1A). This is a situation where the 5′ and 3′ ends of an mRNA are brought in close proximity. The 5′ mRNA cap is bound by EIF4E and the 3′ poly (A) tail is bound by PABP. These two proteins are bridged by the large scaffolding protein EIF4G. To initiate translation, the ternary complex, composed of EIF2:tRNA_i_^Met^:GTP facilitates the decoding of the start codon, which results in GTP-to-GDP hydrolysis. In response to stress, translation is rapidly inhibited at this initiation step, which, in most cases, results in the induction of SGs formation and the storage of EIFs. The inhibition of two translation pathways can induce the formation of SGs [20]: the phosphorylation of a subunit of EIF2, EIF2α (or EIF2S1) [9], prevents the formation of the translation initiation complex EIF2α-tRNA^met^-GTP (Figure 1A,B), and the dephosphorylation of 4EBP that inhibits mRNA circularization, so as to impair re-initiation at the start codon (Figure 1C) [35,36]. These two pathways are not mutually exclusive. In the current state of the literature, most of the investigated stressors induce the formation of SGs via the phosphorylation of EIF2α. However, one or both pathways could be activated, depending on the type of stress [9,37]. While it is intuitive to think that cells shut down translation to preserve energy, one can wonder what would be the other consequences of SGs formation in response to environmental stress.

## 3. Stress Granules Are Pro-Survival Entities at the Cytoplasmic Level

Stressors triggering the formation of SGs can be as diverse as extreme temperatures (hot or cold), oxidative stress, osmotic stress, endoplasmic reticulum (ER) stress, mitochondrial stress, or UV irradiation (previously reviewed [30,31]). Several lines of evidence point toward pro-survival benefits of SGs formation, possibly explaining the evolutionary conservation of this process. Indeed, the mutation or complete knock-out of specific proteins that are involved in SGs formation, or treatments that impair SGs formation, results in rapid cell death after stress exposure [38,39,40,41,42,43,44]. This pro-survival effect of SGs formation could be explained by several independent mechanisms:First, many pro-apoptotic signaling molecules are sequestered in SGs and it has been proposed that it prevents them from activating the pro-apoptotic cascade. This is the case for RACK1 (Receptor of Activated Protein C Kinase 1), TRAF2 (TNF receptor-associated factor 2), and RSK2 (Ribosomal S6 kinase 2) [45,46,47].Second, while not fully characterized, SGs protect cells from oxidative insults by reducing the level of cellular reactive oxygen species (ROS) [39,41,48]. Indeed, when the expression level of a major SGs regulator G3BP1 (RAS GTPase-activating Protein-Binding Protein 1) drops, ROS generation after exposure to oxidative insult increases. Moreover, the overexpression of G3BP1 reduces the level of ROS as compared to wild type cells. Cells expressing a truncated form of the protein that abrogates SGs formation have increased ROS production. Similar results have been obtained with USP10 (Ubiquitin carboxyl-terminal hydrolase 10), another SGs regulating protein [39].Third, SGs formation reduces the cellular energetic needs during stress by restricting the process of translation, which consumes a lot of ATP. By protecting mRNAs from stress-induced degradation, this allows cells to restart translation as soon as the stress is resolved without having to synthetize de novo RNAs [49]. Additionally, SGs sequester untranslated mRNAs concomitantly with the global inhibition of translation [17]. Some mRNAs, such as chaperone mRNAs, are excluded from the SGs structures, so that they can be preferentially translated during the time of the stress and participate in the proper protein folding and avoid functional defects [17,19,50]. By these actions, SGs are described as triage centers for translation of mRNAs during stress exposure [17]. One leading hypothesis is that SGs are able to reshape translational patterns under stress exposure [19,51].

## 4. Stress Granules Are Involved in Cancer Progression

In some cases, SGs pro-survival role might not be beneficial for the host, such as in the context of cancer. Cancer cells are frequently, if not systematically, exposed to stresses, such as hypoxia and nutrient deprivation, two stressors able to induce the formation of SGs [10] and promote resistance to therapies, which suggest a pro-survival role of SGs in this context [18]. Chemotherapies (CT) can also induce SGs formation [11,12,14,15,16,52]. The cancer cell capacity to form SGs in response to CT is correlated with cell survival in vitro [11,12,14,15,16,52]. Blocking the induction of these chemotherapy-induced granules by interfering with the phosphorylation of EIF2α increases the efficiency of CT [16,53]. Along the same lines, some molecules can prevent SGs formation and restore sensitivity to CT. This was shown in a study using hypoxia to induce chemo-resistance in HeLa cells. A screen of small molecules revealed that β-estradiol, progesterone and stanolone prevent SGs formation and restore the sensitivity to CT in HeLa cells [18]. The same molecules used in the MCF7 breast cancer cell line did not block SGs formation, or the chemoresistance induction, suggesting a cell-specific effect of these molecules on SGs [16,18]. The ability to block SGs formation seems to vary from patient-to-patient, just as treatment response. Overall, this study suggests that blocking SGs formation in cancer may be an interesting option against cancer cells, provided that a compound targeting SGs and able to overcome the patient/tissue heterogeneity could be found [18].

During cancer development and progression, tumor cells acquire driver mutations that are responsible for cell transformation, then progression, and aggressiveness of the disease. Malignant transformation is a complex and multifactorial mechanism, involving major changes in the genome, and in transcription and translation programs of the cell. For example, Epithelial-to-Mesenchymal Transition (EMT), acquisition of stemness features, or the acquisition of drug resistance involve specific modifications of translation programs. A growing tumor is an extremely dynamic environment where stressors, such as mechanical constriction, hypoxia and/or starvation (nutrient and/or glucose) play a role at multiple levels. SGs could play a critical role in integrating these stressors into changes in translation that leads to cancer progression [18,51]. Recent studies have demonstrated that such changes occur after exposure to hypoxia stress [18,54,55,56].

## 5. SGs Regulators and Their Role in Cancer

SGs are composed of numerous proteins [31] and the number of regulators increases with the expansion of the field. As a proof of concept, here we focus here on the proteins that are the most used as SGs markers: TIA-1 (T-cell-restricted intracellular antigen-1), TIAR (TIA-1-related protein), G3BP1 (RAS GTPase-activating protein-binding protein 1), and G3BP2 (RAS GTPase-activating protein-binding protein 2) [31]. We also looked at two robust regulators of SGs aggregation: the SGs formation enhancer CAPRIN-1 (Cell Cycle associated protein 1) and the inhibitor of SGs formation USP10 (Ubiquitin carboxyl-terminal hydrolase 10) [57] (Figure 2).

TIA-1 and TIAR have documented roles in immunity, RNA splicing, and translation. Structurally, they bind RNA through RNA Recognition Motifs (RRM) (Figure 2). They are the historical markers for SGs [4,31]. Their overexpression induces the spontaneous formation of SGs [4], and their individual knock-out reduces the cell ability to SGs formation in MEF (mouse embryonic fibroblast) in response to oxidative and thermal stress [58]. Surprisingly, the depletion of TIAR using a doxycyclin-inducible system triggers stress by activation of PKR and SGs induction in 50% of the HEK293 cells [59]. This discrepancy could be explained by the difference in the cell line used and the way to inhibit expression. The knockout is a stable and permanent protein depletion, whereas the doxycyclin system induces a sudden change in the cell environment that could be perceived as a stress for the cell. The role of TIA-1 and TIAR in cancer is still debated, because, in some studies, their depletion accelerates mitotic entry and proliferation [60,61], whereas, in others, they contribute to angiogenesis, tumor growth, and chemoresistance [62]. This discrepancy could be explained by post-translational modifications. For example, in basal conditions, expression of TIA-1 enhances SGs formation, but TIA-1 oxidation results in the opposite effect [41].

The G3BP1 protein also contains an RRM (Figure 2) and has been reported to have helicase and RNAse activity under normal conditions [22,58]. G3BP1 is closely related to another protein, G3BP2, with which it shares 98% identity. G3BP1 and G3BP2 are currently considered as the master regulators of SGs. Their overexpression also induces spontaneous formation of SGs [22,63]. Individual knockout partially inhibits or delays the formation of SGs in response to oxidative, ER or mitochondrial stress [49,57], but the double knock out completely abolishes the formation of SGs in response to oxidative, ER, mitochondrial stress, as well as EIF2α-independent stress [57]. G3BP1/2 have been related to tumor initiation [64], proliferation, [65,66,67], migration [66,67,68], and invasion [66,67] of tumor cells. Those functions are attributed to the RRM and the NTF2 protein domains [65,66], which are also essential for SGs formation [22]. G3BP1 and G3BP2 have been linked to chemo- and radio-resistance [69,70], as their depletions in cell lines increase treatment efficiency. Ultimately, two studies investigated the role of G3BP1 in cancer development in in vivo experiments. In the first one, hepatocellular carcinoma cells were injected into the tail vein to mimic metastatic dissemination by lung colonization [68]. The downregulation of G3BP1 decreased lung colonization, whereas the overexpression of G3BP1 increased lung colonization [68]. In the second study, orthotopic tumor xenografts from ACHN renal carcinoma cell were used and the primary tumor site formation as well as metastasis in the lung and liver were monitored. G3BP1 depletion strongly inhibited both the primary site and the metastasis formation [67].

Finally, USP10 and CAPRIN-1 are two interactors of G3BP1 (Figure 2) that compete with each other to interact with their target [57]. Both bind G3BP1 on a short linear motif Phe-Gly-Asp-Phe (FGDF-motif), and they have opposing effects on SGs formation: CAPRIN-1:G3BP1/2 interaction favors SGs formation, whereas USP10:G3BP1/2 interaction inhibits their formation [57]. CAPRIN-1 overexpression is also reported to induce spontaneous formation of SGs and its inhibition decreased SGs formation in response to oxidative and ER stress as well as in response to EIF2α-independent stress [57]. In contrast, the overexpression of USP10 inhibits SGs formation in response to oxidative, mitochondrial, and EIF2α-independent stress. USP10 depletion does not significantly impact SGs formation in response to oxidative, mitochondrial and EIF2α-independent stress [57]. Those two proteins are not the sole regulators of G3BP1 aggregation, because the removal of the FGDF-motif does not influence the formation of SGs. CAPRIN-1 and USP10 not only have opposite effect on SGs regulation, but also have opposed incidence on cell proliferation and tumor development. CAPRIN-1 enhances proliferation [71,72,73], whereas USP10 inhibits tumor progression and invasion [74,75].

## 6. SGs Proteins as Prognostic Markers: At the mRNA or Protein Level?

SGs proteins regulator expression level can be used to estimate the capacity of cells composing a tissue to form SGs [22,57,58,76]. In line with this, we analyzed publicly available gene expression data of breast [77,78,79,80,81,82,83,84,85,86,87,88,89,90,91,92,93,94,95,96,97,98,99,100,101,102,103,104,105,106,107,108,109,110,111], colon [112,113,114,115,116,117,118,119,120,121], and pancreatic [122,123,124,125,126,127,128,129,130,131,132,133,134,135] cancers as a proof of concept. mRNA levels for *G3BP1*, *TIA-1*, *TIAR*, and *CAPRIN-1* are mostly upregulated (*p* ≤ 1.07 × 10^−4^) in primary tumors when compared to healthy tissues (Appendix A). This correlates with observations made in several patient cohorts, where SGs proteins were upregulated in tumor as compared to healthy tissues [64,65,68,136,137,138,139,140,141,142,143,144,145,146,147,148,149,150,151,152]. Even if other possibilities exist to explain this phenomenon, we can hypothesize that tumor cells divert and exacerbate a pro-survival mechanism potentially based on SGs, in order to facilitate their survival in response to the numerous stresses encountered. This is true for most primary solid tumors. *CAPRIN-1* and *G3PB1* mRNA show the most noticeable/prominent upregulation in primary tumors (Appendix A).

The difference in expression between healthy tissues and metastases was not as pronounced as between healthy and primary tumors (Appendix A). For breast and colon cancer, mRNAs encoding SGs proteins level between metastases and primary tumors stay steady or decrease. This might be explained by the fact that tumor cells have already adapted their growth to major stresses or because it involves other regulatory mechanisms for SGs formation in breast an in colon cancer. From the three cancers studied here, pancreatic cancer is the most aggressive and the one with lower survival rate (9% five-year survival, as compared to 91% for breast cancer and 63% for colon cancer according to the American Society of Clinical Oncology). It is also the only one where mRNAs encoding SGs proteins are still increasing between primary tumors and metastatic sites, suggesting that cancer cells are still evolving/adapting toward a more aggressive disease.

Despite the fact that an increased expression of mRNA encoding for SGs proteins was a marker of tumor transformation (Appendix A), hazard ratio analyses revealed few significant correlations with patients’ survival (Figure 3). This was not consistent with data from the literature showing that SGs proteins upregulation and poor prognosis are often correlated, in multiple cancer types. Indeed, G3BP1/2^high^ proteins expression is linked to poor prognosis in various tumors, including sarcomas [138], breast [136,137], lung [139], stomach [140,141], liver [68], and prostate cancers [142]. High TIA-1 protein levels correlate with poor prognosis in patients with colorectal cancer [143], lymphoma [144], and hepatocellular carcinoma [145]. A high expression of CAPRIN-1 protein correlates with poor prognosis for patients with osteosarcoma [146], and hepatocellular carcinoma [147,148]. Consistent with an inhibitory effect of USP10 on SGs formation, high protein expression correlates with better prognosis in patients with ovarian [150], lung [151], small intestine [152], prostate [142], and stomach carcinoma [149] (Table 1).

## 7. Conclusions

Taken together, the discordant results between mRNA and proteins expression levels regarding the prognostic value suggest (Figure 3, Table 1):An early increased of mRNA level in the pathology achieved by upregulation of transcription or decrease of RNA degradation. This phenomenon could insure an efficient level of the proteins that are involved in SGs function. This basal level is potentially the result of cancer cells subversion of the SGs mechanism to ensure their survival and it is present in the majority of cancer patients (Figure 4).A post-transcriptional regulation impacting the final level of proteins. Increasing the translation rate of the mRNA encoding a SGs protein would be more efficient than producing more mRNAs. This could be done by specific signal in the non-translated region of the mRNA such as AU-rich element (ARE) [153] or to the length of the mRNA as it was already described [154]. Favoring translation over transcription is perhaps also a mechanism allowing an economy of energy in cancer cells (Figure 4).Or a post-translational modification enhancing the half-life of the protein. The degradation of the protein is delayed and it enhances the overall level of protein (Figure 4).

## 8. Perspectives

The cancer field is still facing the challenge of metastasis and chemoresistance during the disease evolution, despite global survival improvement of cancer patients over the years. Of course, the search for molecular alterations have been successful in the past, contributing to improved tumor classification and the development of efficient targeted therapies. However, some patients do not respond or become refractory to given therapies, meaning that finding treatments that are able to counter drug-induced mechanisms of resistance is the challenge to address. One reason that has limited progress in this field so far is the fact that mechanisms of resistance might only be present during the course of the treatment, and may recess once therapy is stopped. The reversibility of the phenomenon, also termed “plasticity”, makes it difficult to apprehend, since it may not be identifiable in the resected specimen [2]. The mechanisms involved are most probably fast, “acute” and involve “ready-to-use” components, like those of the stress pathways. In this line, the cancer field is accumulating evidence for a role for SGs proteins in the adaptation and survival of cancer cells during tumor growth [64,65,136] and chemoresistance [41,42,43,44,45,46]. All of the SGs proteins that are described in this review have been previously reported to have a role in cancer development and/or cell cycle regulation [60,61,62,64,65,66,67,68,71,72,73,74,75]. These studies collectively suggest that tumor cells have an increased ability to form SGs when compared to non-pathological cells based on their increased expression of SG-related proteins. In addition, prognostic analyses correlate this SGs protein expression with poor survival in patients (Figure 4). The exact mechanism for this increased protein level without transcriptional change is not known yet and could be the result of increased specific translation or decreased protein degradation. Currently, we have no way to distinguish between the roles of these proteins by themselves or as part of the SGs. However, we noticed that all SGs nucleators (TIA-1, TIAR, G3BP1, G3BP2, and CAPRIN-1) have an “enhancing” effect on cancer development properties [62,64,65,66,67,68,71,72,73], whereas USP10, a SGs inhibitor, has anti-cancer properties [74,75] (Figure 5). There are some interesting correlation between other SGs markers and development, such as eIF3s and eIF4s subunits and complexes, UBAP2L or YB1, for example [138,155,156,157,158,159,160]. All of those proteins have, in common, to be recruited to SGs or to have role in their regulation [31,138,161,162]. This collection of evidences points toward a role of SGs in cancer. Meanwhile in vivo experimental evidences supporting a role for SGs in cancer development are accumulating. Limiting the formation of SGs through genetic [67,68] or chemical inhibition [53,163] showed promising results on the inhibition of primary and metastasis formation. Of course, we cannot exclude that SGs proteins have additional and complementary roles to SGs in tumorigenesis. SGs have already been observed in vivo [26,51,164] and the presence of SG-like foci has been reported in in vivo mouse xenografted tumors [138]. Recently, the field is starting to observe SGs in clinical samples from patients with pancreatic adenocarcinoma [165]. While promising, SGs observations in patients are still at the leading edge and need to be reproduced in other kinds of cancer. Overall, SGs might thus turn out to be important actors of tumor cells plasticity in response to the various stresses encountered, which will make them a target of choice in the fight against tumor development, progression, and prevention of chemoresistance. Further studies are warranted in order to understand SGs the mechanism of action in the development of aggressive and invasive cancers and how to block them to improve patient care, especially for refractory patients or in the prevention of chemoresistance.

## Figures and Tables

**Figure 1 cancers-12-02470-f001:**
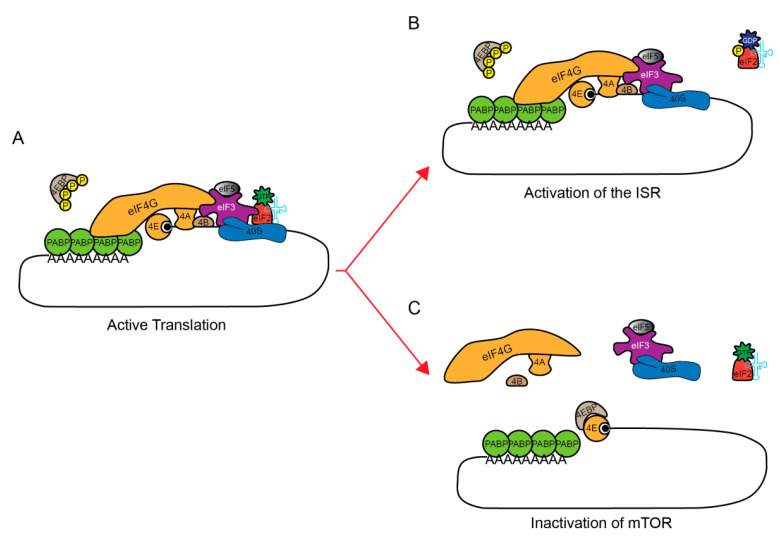
Inhibition of translation pathways to induce the formation of SGs. (**A**) Active translation initiation complex (basal condition). Two pathways could be activated to induce translation inhibition upstream SGs formation. Under basal conditions, EIF2α is not phosphorylated and could allow the formation of the EIF2: tRNA_i_^Met^: GTP ternary complex of translation initiation. Additionally, MTOR is active and constitutively phosphorylates EIF4E-Binding Protein (4EBP). (**B**) The phosphorylation of a subunit of EIF2, EIF2α (or EIF2S1), by one (or more) kinase(s), notably HRI/EIF2AK1, PKR/EIF2AK2, PERK/EIF2AK3 and/or GCN2/EIF2AK4, prevents the hydrolyzed GDP from leaving the ternary complex EIF2α-tRNA^met^-GTP by blocking the formation of an active complex with ATP necessary for translation initiation. (**C**) Hyper-phosphorylated 4EBP cannot interact with EIF4E, the mRNA cap-binding protein. However, induction of a stress response inactivates MTOR leading to a rapid de-phosphorylation of 4EBP, thereby allowing it to interact with EIF4E. The EIF4E: 4EBP interaction prevents EIF4E: EIF4G complex formation.

**Figure 2 cancers-12-02470-f002:**
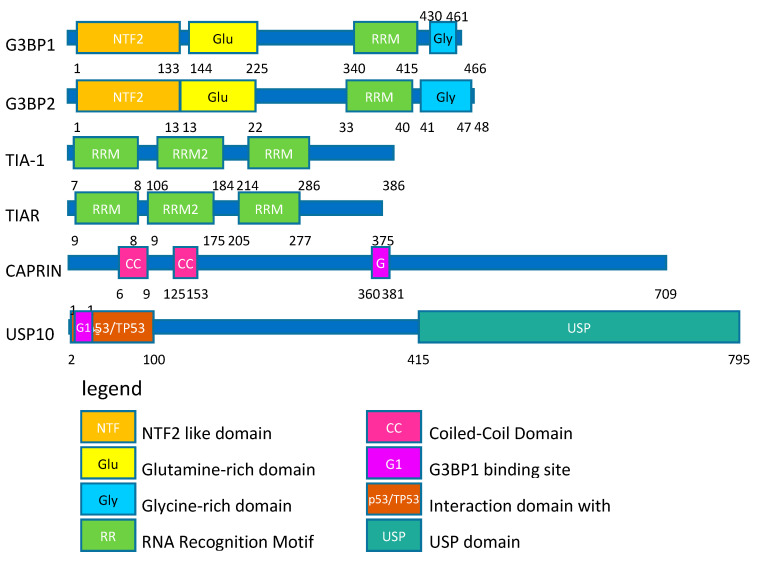
Structure of stress granules (SGs) related proteins. SGs related proteins are represented according to domains described in the Uniprot database.

**Figure 3 cancers-12-02470-f003:**
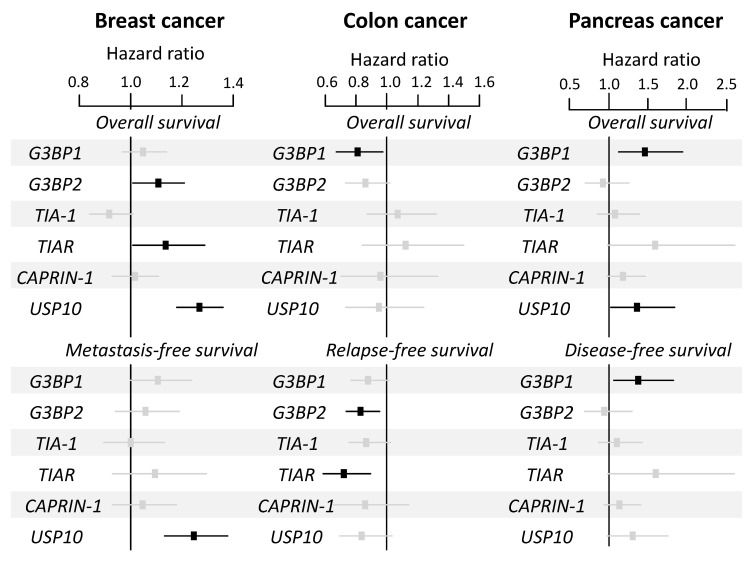
Forest plot showing the hazard ratio for survival events of the mRNA expression of SGs genes according to overall and metastasis-free survival in breast cancer, overall and relapse-free survival in colon cancer, and overall and disease-free survival in pancreatic cancer patients. A ratio greater than one indicates a poor prognosis and a ratio lower than one indicates good prognosis. The black squares correspond to significant genes and the grey ones to non-significant genes. The analysis was performed according to Appendix A.

**Figure 4 cancers-12-02470-f004:**
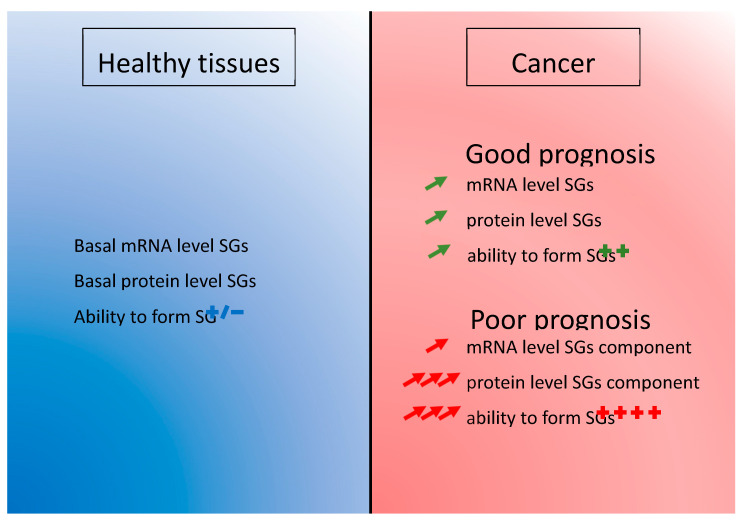
The two steps regulation of SGs component in cancer. During oncogenesis, there is a global increased expression of mRNA encoding SGs-related proteins. Protein, but not mRNA, expression levels are prognostic for survival, suggesting different layers of regulation for SGs in cancer.

**Figure 5 cancers-12-02470-f005:**
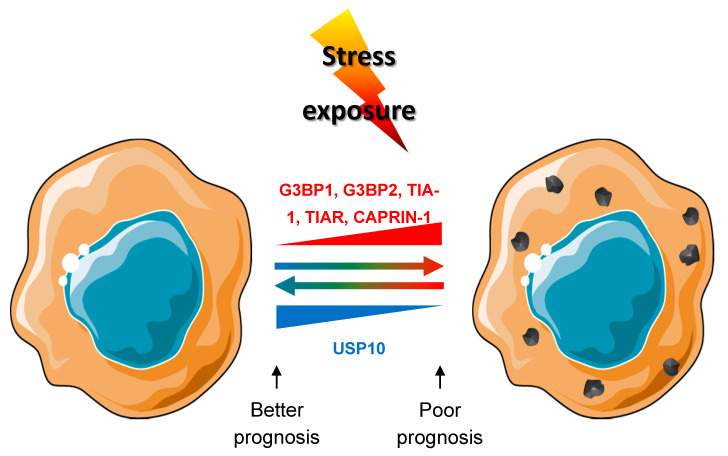
Link between SGs proteins and prognosis in patients. Overexpression of G3BP1, G3BP2, Table 1. TIAR and CAPRIN1 are linked to increased ability to form SG. Whereas increased expression of USP10 decreases the cell ability to form SGs in response to stimuli. Increased expression G3BP1, G3BP2, TIA-1, TIAR and CAPRIN1 proteins are reported as linked to poor prognosis, whereas increased expression of USP10 is a good prognosis for patients.

**Table 1 cancers-12-02470-t001:** Kaplan–Maier analysis on SGs protein (high vs. low expression). Summary of Kaplan–Meier survival analysis found in the literature [64,65,68,136,137,138,139,140,141,142,143,144,145,146,147,148,149,150,151,152]. Data are presented for patients with high expression of the protein compared with patient with low expression of the protein. «Poor» means that patient with high protein expression have a worse prognosis than patients with a lower expression. «Good» means that patients with high protein expression have a better prognosis than patients with a lower expression.

Cancer Type	G3BP1	G3BP2	TIA-1	CAPRIN-1	USP10
Breast	Poor [136]	Poor [137]			
Colon/Colo-Rectal			Poor [143]		
Sarcoma	Poor [138]				
Stomach	Poor [140,141]				Good [149]
Lung	Poor [139]				Good [151]
Liver	Poor [68]		Poor [145]	Poor [147,148]	
Prostate		Poor [142]			Good [142]
Intestinal					Good [152]
Ovarian					Good [150]
Osteosarcoma				Poor [146]	
Lymphoma			Poor [144]

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
