# Peer review of "Revisiting the Concept of Stress in the Prognosis of Solid Tumors: A Role for Stress Granules Proteins?"

_cancers, 2020, doi:10.3390/cancers12092470_

Round 1

Reviewer 1 Report

The way describing the formation of SG is some-how not up to date. The concept that SG form in a phase-separation process was not discussed and this is relevant in cancer. The role of mTOR pathway in inducing SG formation is specific to certain type de stress which should be specified here as mTOR activity is not required for SG formation in most type of stress. The role of eIF4E, eIF4G, eIF4A…in SG formation could have been described as most these factors play a crucial role in tumor progression and possibly in chemoresistance.  

Author Response

This manuscript is a resubmission of an earlier submission. The following is a list of the peer review reports and author responses from that submission.

Round 1

Reviewer 1 Report

This review presents correlations between the expression of selected SG regulators and pronostic in cancer patients. However, due to the lack of in vivo (e.g. using tumors) experimental evidence, it is very preliminary to document a role of SG formation in tumorigenesis. There is also no evidence that the role of selected proteins in cancer is mediated by their ability to activate SG pathway. This review lacks accurate description of the role of selected proteins in cancer and how this role can be uncoupled from SG regulation.

Other points (Reviewer comments are underlined:

1-‘’In 2015, an inventory of the literature mentioned more than a hundred of proteins known to be recruited to SGs’’. This statement needs the corresponding reference

2-‘’In homeostatic conditions, active translation is facilitated by the formation of a closed-loop mRNA’’. This concept was recently challenged in Adivarahan et al Mol Cell 2018. Please include this reference.

3-‘’In response to stress, translation is rapidly inhibited, which, in most cases, results in SG formation’’ Please specify that SG formation occurs when translation is inhibited at its initiation phase.

4-‘’The inhibition of two translation pathways can induce the formation of SGs (the P-eIF2a and 4E-BP1). Inactivation of eIF4A was also shown to induce SG (Mazroui et al; 2006; Tauber et al  Cell 2020). Please include the eIF4A data

5-‘’A screen of small molecules 138 revealed that β-estradiol, progesterone and stanolone prevent SGs formation and restore the sensibility to CT’’. This statement needs the corresponding reference.

6-‘’The same molecules used in the MCF7 cancer cell line did not block SG formation nor the chemoresistance induction’’. Please discuss the discrepancy between points 5 and 6.

7-The authors mentioned that both overexpression of TIAR and its depletion induce SG? Please discuss these conflicting results.

8-The authors mentioned that depletion of specific proteins abolished SG formation. It is important to specify the stress conditions used to assess SG formation in depleted cells

9-Need to explain why the focus is on G3BP1, TIA1, TIAR and CAPRIN1. Other SG components and regulators were also shown to be involved in cancer.

10-“A post-transcriptional regulation; impacting the final level of proteins. Increasing the translation rate of the mRNA encoding a SG protein would be more efficient than producing more mRNAs. Favoring translation over transcription is perhaps also a mechanism allowing an economy of energy in cancer cells’’. Could the authors propose models to explain mechanisms that favor translation of mRNAs encoding SG regulators? In other word, how cancer cell select those mRNAs for translation?

11-Table 3: The authors presented data on G3BP, G3BP1 and G3BP2? Is this mean that they are 3 G3BP isoforms?

Reviewer 2 Report

In this review the authors summarized the role of Stress Granules (SG) in the solid tumor prognosis. The topic is novel and interesting. However, the writing is not clear enough and not well focused. The authors should try to list the results in a more logic and consequent way. The writing often suffers for lack of appropriate references that are often missing. The point of view of the authors is also missing. Making a list of results does not mean that the authors wrote a review. They should also include critical aspects of the topic and its clinical application.

- In lane 28: “UV, tissue remodeling…”, instead of dots “…” I would write “etc."

- In the Introduction references are missing for the main topics that are introduced, including, heat shock response, unfolded protein response, cell response to stress (survival or apoptosis). Many interesting reviews have been written recently about these topic and the authors should cite the most recent ones. Please amend.

- heat shock response or unfolded protein response could also be abbreviated as “HSP” and “UPR”.

- In lane 39: “….the formation of Stress Granules (SGs) [1]”, perhaps a more appropriate reference should be included. For instance: “Reineke LC, Neilson JR. Differences between acute and chronic stress granules, and how these differences may impact function in human disease. Biochem Pharmacol 2019, 162: 123-131”. Please amend.

- Lane 45: “are membrane less cytoplasmic condensates”, what does it mean? Perhaps the autors mean “membrane free cytoplasmic condensate”. Please check and amend.

-Lane 87. 3 Paragraph: This paragraphs looks superficial, perhaps the authors could better focus it and also insert a Table or a scheme to let the reader understand the role of SG in pathological conditions. Please amend.

- Same comment for paragraph 4. Please try to better focus the meaning of the results.
